# Development of an artificial intelligence system to identify hypoglycaemia via ECG in adults with type 1 diabetes: protocol for data collection under controlled and free-living conditions

Owain Cisuelo [ID],[1] Katy Stokes [ID],[1] Iyabosola B Oronti,[1] Muhammad Salman Haleem [ID],[1] Thomas M Barber,[2,3,4] Martin O Weickert,[2,3] Leandro Pecchia,[1,5] John Hattersley[1,4]

For numbered affiliations see end of article.

**Correspondence to**
Owain Cisuelo;
owain.cisuelo@warwick.ac.uk

## ABSTRACT

**Introduction** Hypoglycaemia is a harmful potential complication in people with type 1 diabetes mellitus (T1DM) and can be exacerbated in patients receiving treatment, such as insulin therapies, by the very interventions aiming to achieve optimal blood glucose levels. Symptoms can vary greatly, including, but not limited to, trembling, palpitations, sweating, dry mouth, confusion, seizures, coma, brain damage or even death if untreated. A pilot study with healthy (euglycaemic) participants previously demonstrated that hypoglycaemia can be detected non-invasively with artificial intelligence (AI) using physiological signals obtained from wearable sensors. This protocol provides a methodological description of an observational study for obtaining physiological data from people with T1DM. The aim of this work is to further improve the previously developed AI model and validate its performance for glycaemic event detection in people with T1DM. Such a model could be suitable for integrating into a continuous, non-invasive, glucose monitoring system, contributing towards improving surveillance and management of blood glucose for people with diabetes.

**Methods and analysis** This observational study aims to recruit 30 patients with T1DM from a diabetes outpatient clinic at the University Hospital Coventry and Warwickshire for a two-phase study. The first phase involves attending an inpatient protocol for up to 36 hours in a calorimetry room under controlled conditions, followed by a phase of free-living, for up to 3 days, in which participants will go about their normal daily activities unrestricted. Throughout the study, the participants will wear wearable sensors to measure and record physiological signals (eg, ECG and continuous glucose monitor). Data collected will be used to develop and validate an AI model using state-of-the-art deep learning methods.

**Ethics and dissemination** This study has received ethical approval from National Research Ethics Service (ref: 17/NW/0277). The findings will be disseminated via peer-reviewed journals and presented at scientific conferences.

**Trial registration number** NCT05461144.

### STRENGTHS AND LIMITATIONS OF THIS STUDY

⇒ This two-phase study will obtain data in both the controlled environment of a metabolic chamber and in free-living conditions, overcoming limitations of previous studies limited to controlled conditions.
⇒ The second phase of the protocol has been designed with non-compliance in mind in order to determine how a model developed in controlled settings works with real-world data.
⇒ This is a single-centre observational study aiming to recruit from the local population without discrimination of characteristics such as sex, ethnicity or race, which may limit the generalisability of the results.

## INTRODUCTION

Type 1 diabetes mellitus (T1DM) is a chronic disorder caused by autoimmune damage to the insulin-producing pancreatic beta cells, leading to elevated blood glucose concentration (hyperglycaemia).[1] T1DM requires exogenous insulin treatment,[2] of which hypoglycaemia is a potential side effect.[3] The prevalence of T1DM globally is 5.9 per 10 000 with the incidence of 15 per 100 000 people.[4 5] In the UK, the prevalence is reported as fifth highest with 8.6%, and incidence of T1DM is rising 5% every year.[5] There is no cure for diabetes; therefore, effective management can be pivotal in reducing risk of adverse events as well as delay the onset of long-term complications.[6 7]

The self-monitoring of blood glucose (SMBG) can play a key role in effective management of diabetes. SMBG can empower patients to adhere to lifestyle interventions, such as changes to diet and exercise regimen, as they have access to immediate feedback on the effects.[8] Active participation

from the patient in their care has been shown to improve outcomes such as reducing the risk of diabetic complications.[9] Current methods for glucose monitoring have several limitations. The most common method for SMBG is an invasive 'finger-prick test' in which a small sample of blood is collected from the fingertip and analysed by a handheld device called a glucometer. This invasive procedure has several documented barriers preventing SMBG such as inconvenience, cost and discomfort.[10] Moreover, the devices and associated consumables are prone to error if not operated correctly,[11] requiring repeat attempts and generating unnecessary waste. Hypoglycaemia typically occurs during sleep,[12] and those receiving insulin therapy are at increased risk due to a combination of factors such as impaired counter-regulatory hormone response and warning symptoms going unobserved.[13] This presents an obvious limitation to the effectiveness of an invasive 'finger-prick test' as a tool for SMBG. An alternative to 'finger-pricking' are continuous glucose monitoring (CGM) devices, which require a small cannula inserted in the top layer of skin to continuously measure glucose in the fluid between the cells. They can be described as minimally invasive as after the initial fitting, they are routinely worn continuously for up to 14 days. However, despite being shown to be a beneficial tool for SMBG, national clinical guidelines for the UK do not recommend routine use of CGM devices for the management of T1DM due to insufficient evidence demonstrating their efficacy and cost-effectiveness.[14]

It has been demonstrated, by this group[15 16] and others,[17–19] that glycaemic events can affect certain cardiac characteristics, which can be extracted from readings of the electrical activity of the heart, the ECG. With increasing availability of wearable devices for tracking physiological signals, studies have been undertaken to investigate prediction of blood glucose concentration or glycaemic event detection using artificial intelligence (AI), by combining data or features extracted from the ECG with blood glucose measurements. Such an approach would address the need for non-invasive continuous blood glucose monitoring. Studies have shown promising predictive performance based on data collected from healthy subjects[15 16 20–22] or adolescent patients with T1DM.[19 23] In a review of techniques for detecting hypoglycaemia, Diouri et al note that the use of ECG-based techniques has only been investigated in small cohort trials, and that the success of such approaches relies on further, ideally larger, trials and validation in patients with diabetes and cardiac diseases.[24]

This protocol expands on an initial pilot, results of which are published,[15] which made opportune use of applicable secondary data collected for a separate study[25] on healthy elderly men. The aim of the pilot was to use the data to create a personalised AI model for the detection of nocturnal hypoglycaemia. The developed and validated deep learning model achieved 90% for specificity (ability to identify true positives) and sensitivity (ability to identify true negatives), which are clinically relevant metrics in

relation to diagnosis of disease.[26] The aim of this protocol is to build on this initial work, by obtaining new data to validate the technique for people with diabetes and test its applicability beyond controlled conditions, to that of normal living conditions.

## Study objectives

The purpose of this observational study is to acquire continuous physiological data from adults with T1DM over a period of several days. Data obtained from wearable sensors and recorded in diaries to track lifestyle activities will be used to carry out our primary and secondary objectives.

### Primary objective

The primary objective of this study is to further develop and validate in a new population (T1DM) our previously developed AI approaches for non-invasive hypoglycaemia detection. We will extract and analyse ECG-derived features from the raw signal and determine their relationship with the glycaemic status. State-of-the-art algorithms will be used to create a mathematical mapping from the ECG-derived features to the glycaemic status.

### Secondary objective

To examine the impact of physical activity and diet on glycaemic events and incorporate relevant features into the model.

## METHODS AND ANALYSIS
## Study setting

This two-phase observational study will be conducted at the Human Metabolism Research Unit (HMRU) and in free-living conditions. During phase 1, the participants attend an inpatient protocol at University Hospital Coventry and Warwickshire (UHCW) in a calorimetry room for up to 36 hours. The calorimetry room is a controlled environment, which allows for precise regulation of parameters such as ambient temperature, air pressure, humidity and subject behaviour enabling us to obtain high-quality baseline ECG and glucose data from the wearable sensors. In phase 2, the same sensors will be used to collect data under free-living conditions, for a period of up to 3 days, during which the participants are free to go about their normal daily activities without restriction. Participants will be asked to complete diaries for daily activity, food intake and sleep.

### Inclusion and exclusion criteria

The study will be open to all adult individuals living independently with T1DM who are comfortable with a stay in a calorimetry room. The inclusion and exclusion criteria are summarised in box 1.

### Enrolment procedure

Participants will be recruited from the Warwickshire Institute for the Study of Diabetes, Endocrinology, and Metabolism Clinic at UHCW. We aim to recruit up to 30

## Box 1 Inclusion and exclusion criteria

**Inclusion criteria**
⇒ Aged 18 years or older.
⇒ Without acute illness or ongoing clinical investigation.
⇒ Participants with an ongoing medical condition will only be included after detailed consultation with clinical and dietetics members of the team.

**Exclusion criteria**
⇒ Children (under 18 years).
⇒ Any adult who lacks decisional capacity.
⇒ Claustrophobic and/or isolophobic patients, or those with needle phobia.
⇒ Individuals who have undertaken recent abnormal exercise, radiation exposure within the preceding 24 hours of entering the whole-body calorimeter and feeling unwell in any way.
⇒ Any medical/endocrine problem that could affect energy expenditure (eg, thyroid problems, Cushing's syndrome).
⇒ Chronic inflammatory disorders like rheumatoid arthritis, or long-term use of steroids or other immunomodulators like ciclosporin, azathioprine and beta blockers.
⇒ Currently actively losing weight.
⇒ Depression or any psychiatric illness.

adult patients with T1DM who may enrol in either or both phases of the study. The process of identifying potential participants began in September 2022 and the first session took place in December 2022. Participant recruitment will be an ongoing process, expected to conclude in September 2023. All participants will receive a participant information sheet and written informed consent will be obtained.

### Sample size

As this is an observational study, an exact sample size calculation is not possible. A target of 30 participants will be enrolled, based on a pragmatic approach, which ensures sufficient data are available for analysis. Full participation in the protocol will yield 108 hours of continuous data per participant. We aim to capture the daily blood glucose fluctuations known as glycaemic variability,[27] a feature of impaired glucose metabolism. However, it is not possible to anticipate how many hypoglycaemic episodes will be recorded. ECG data, labelled with corresponding glucose concentration, will be analysed in excerpts of variable length from individual heartbeats (cardiac cycles) to several minutes, resulting in a large sample size.

### Study protocol
#### Calorimetry room

The participant will stay in the calorimetry room for up to 36 hours, during which there will be set times for meals, light exercise and rest. Meals will be provided with known total energy and macronutrient content. Water will be provided ad libitum. While in the calorimetry room, wearable sensors will record physiological data continuously.

Other activities will be performed during this time, specifically: (a) up to 12 serial blood samples of 30 mL will be obtained from a peripheral venous catheter inserted at the beginning of phase 1 to measure venous glucose and insulin concentration, (b) finger-prick blood samples will be taken periodically for capillary glucose measurements, (c) up to 12 saliva samples will be taken to analyse salivary concentration of cortisol and melatonin, (d) all urine will be collected as voided for protein oxidation analysis, (e) movement and activity will be assessed by motion sensors, (f) simple questionnaires will be used to measure food intake and activity as well as subjective aspects such as appetite, satiety and wellness, and (g) daytime blood pressure will be monitored using an ambulatory blood pressure monitor (hourly readings).

#### Free-living

While free-living, the participant will continue to wear the monitoring devices to record ECG and glucose concentration. If tolerated, participants will also be fitted with an ambulatory blood pressure monitor for the first 24 hours of the free-living phase. Instructions will be provided on how to affix and operate the devices. During this time, participants will also be required to keep brief diaries detailing physical activity, food intake and sleep. The first is a daily activity diary, to be completed at the end of each day, detailing: any device removal and replacement, physical activity, food intake, alcohol intake and caffeine intake. The second is a standardised sleep diary, the Consensus Sleep Diary-M,[28] to be completed on waking in the morning.

### Devices
#### Continuous glucose monitoring

Continuous glucose levels will be measured using a Free-Style Libre 2 flash glucose monitoring system, which can be worn for up to 14 days. Although the sensor is sampling continuously, the glucose concentration is reported in 15-minute intervals. The sensor can store data for a period of 8 hours. As such, data will be extracted at regular intervals using a smartphone preconfigured with the companion application. The sensor is water resistant and can be used while bathing, showering, swimming or exercising, and is worn on the back of the upper arm. The device has been evaluated for accuracy and user experience and found to have been generally well received by patients with T1DM who used it for 10–14 days.[29]

#### ECG

Medtronic Zephyr BioPatch is a CE-marked device, indicating that the product complies with European Union safety, health and environmental requirements, which operates across one lead within an ECG amplitude range between 0.25 and 15 mV, with a sampling frequency of 250 Hz. The device will be affixed to the wearer's skin via two electrodes placed in the centre of the chest. It can also be worn in a secondary configuration attached to a fabric harness, which is worn across the chest if the electrodes are not tolerated. The Zephyr BioPatch also has additional sensors to record physiological parameters such as activity levels, posture and breathing rate, which

are all reported at 1 Hz.[30] The device will be removed before bathing, showering or swimming, and reaffixed afterwards. The battery is rated for 35 hours of continual use. Each participant will be provided with fully charged devices for the free-living phase of the study.

## Data management

All data collected are owned by the UHCW National Health Service (NHS) Trust. All electronic data, physiological or otherwise, generated as part of this study will be anonymised and stored in the HMRU database on secure servers and backed up and protected in accordance with NHS guidelines. Anonymised data will be made available to researchers for analysis under an Institutional Data Sharing Agreement between the UHCW and University of Warwick. Data will be extracted from the devices, anonymised, stored and made available for analysis when the participant has concluded the protocol.

## Data analysis and modelling

### Data processing

The effect of glycaemic events on the ECG signals will be examined. It is anticipated that ECG signals will be affected by noise and artefacts due to body movement or heavy respiration. Therefore, we will apply preprocessing methods such as baseline wander removal to the raw signal to remove low-frequency noise. We will then identify individual heartbeats in the ECG and detect fiducial points using our ECG segmentation tool,[31] which has been used in a recent study for cardiovascular disease detection.[32] Heartbeats will then be grouped into 15-minute excerpts, corresponding to the sampling of the CGM, and annotated with the glycaemic state (ie, hypoglycaemic, normal, hyperglycaemic) according to the thresholds defined in table 1. Due to the lower sample frequency of the CGM, we will perform linear interpolation to estimate glucose measurements at every second.

### Data analysis

Data will be obtained from several days of continuous physiological monitoring of participants. During analysis, the data will be divided into smaller excerpts such as 1, 2 or 5 min intervals. In addition, beat-level ECG samples will be analysed; therefore, the total number of samples available during analysis is ambiguous.

**Table 1** Glycaemic event thresholds for people with T1DM[51–57]

| Glycaemic event | Blood glucose concentration (mmol/L) |
|---|---|
| Severe hypoglycaemia | <2.8 |
| Hypoglycaemia | >2.8 and <3.9 |
| Euglycaemia | >3.9 and <11.1 |
| Hyperglycaemia | >11.1 and <13.9 |
| Severe hyperglycaemia | >13.9 |

T1DM, type 1 diabetes mellitus.

Characteristics of an ECG can be represented by heart rate variability (HRV) parameters and can be categorised into time-domain features, frequency-domain features and non-linear features.[33] We aim to use physiological signal processing packages such as NeuroKit2[34] and HeartPy[35] to determine these features. Furthermore, we aim to determine ECG beat parameters, which include length and slope among the fiducial points (P, Q, R, S and T). We will perform statistical tests to determine the statistical significance of different HRV features and ECG morphology parameters among different glycaemic values to determine which parameters are associated with hypoglycaemia.

### Covariates

The autonomic nervous system is responsible for maintaining homeostasis and regulates processes such as blood pressure,[36] digestion,[37] metabolism[38] and circadian rhythm,[39] all of which may affect cardiac function and show as changes in the ECG morphology.[40–43] Additionally, exercise can affect the QT interval, that is the ECG section representing ventricular depolarisation, due to exercise-induced autonomic response.[44] During phase 1 of the study, daytime blood pressure will be monitored, and the calorific value and macronutrient composition of meals will be recorded. Additionally, exercise sessions will be scheduled throughout the phase. Throughout the free-living phase, all meals, exercise duration and intensity, and sleep times will be recorded by the participant. The data collected via this protocol enable us to examine how the covariates relate to the outcome of interest, that is, the glycaemic status, by performing statistical tests such as analysis of variance.[45] The recording of mealtimes, exercise and sleep enables the data to be analysed during distinct periods of interest such as post-exercise or postprandial.

### Data modelling

The inherent part of the project is to develop AI models for detecting glycaemic events and inferring them with physical activity and diet. The state-of-the-art AI models vary from traditional machine learning models (such as Support Vector Machines, Decision Trees, etc) to advanced deep learning-based models.[46] Traditional machine learning models have the capability to train static features for the development of explanatory models, whereas deep learning models can train raw ECG signals based on spatial and temporal context. The deep learning models will be built upon earlier work developed for healthy subjects.[15] We aim to develop and validate hypoglycaemia detection models that incorporate lifestyle trends such as activity levels and sleep patterns using metrics provided by the sensors and reported in the diaries. For a model to be clinically useful, it should demonstrate generalisability. Therefore, we will test the model on unseen data using a subject-wise cross-validation approach.[47]

## Patient and public involvement

No formal patient and public involvement (PPI) group was convened for this study; however, phase 1 is a standard protocol used at the HMRU and has been continually developed with PPI for over 10 years. The methodology using a single-lead wearable ECG device and CGM was discussed and reviewed with participants for a previous study investigating the effects of resistance exercise and protein supplementation on sarcopenia in healthy older men (ClinicalTrials.gov Registry: NCT03299972). The PPI consists of informal, non-scripted interviews with members of the public post-study via the UHCW PPI forum. Depending on the results obtained, PPI groups may be approached to discuss non-scientific dissemination.

## DISCUSSION

The primary objective of this observational study is to obtain physiological data from people with T1DM using non-invasive wearable sensors to validate and further develop an AI model for automated detection of hypoglycaemia. The initial model[15] was developed using data from healthy individuals and the results determined that the personalised classifiers based on deep learning algorithms can reliably perform automatic detection of hypoglycaemic events using features extracted from the ECG waveform recorded with wearable devices. This study is designed to improve the performance and robustness of this AI model for use in a population with T1DM. Additionally, as a secondary outcome, this study will examine impact of lifestyle habits such as exercise, diet and sleep as predictors in the model.

The study will add to the body of evidence evaluating AI for use in the detection of glycaemic events (eg, hypoglycaemia) in free-living conditions. Current efforts in the literature have been limited by small sample sizes,[48] healthy subject populations[16] and data originating from highly controlled experimental settings.[49] This study protocol has been developed to address the gaps of the existing literature. As such, the strengths of this study design are that data are collected from participants with diabetes, both in controlled and free-living settings. An additional strength of this protocol is that it has been designed with non-compliance in mind. The first phase will take place in a metabolic chamber where the activities of the participants and the operation of the wearable sensors are prescriptive and supervised. This enables us to obtain high-quality baseline data. The second phase will take place in free-living conditions where the participants go about their normal daily activities unencumbered by the wearable sensors, providing real-life data. During the free-living phase, a level of non-compliance and variance is desirable in order to determine how our analysis and model work with real-world data. Non-compliance with the protocol will not exclude the participant from inclusion in the analysis. Any deviations from the protocol will be noted. Participation in both phases of the study is not a requirement; therefore, we are likely to obtain different number of samples for each participant. We will take a pragmatic approach and work with the data available at the end of the study. All analyses will report the number of samples used and if excerpts of data have been excluded and for what reason. Our planned analysis will segment the continuous physiological data into smaller excerpts of variable length, yielding a sufficient sample size for the training algorithm to discover relevant features in the data.

A potential limitation of this protocol is that during the free-living phase of the study, participants will be required to extract data from the CGM at least once every 8 hours. They will also have to remove and reattach the ECG device when engaging in water-based activities or exchanging the device for a new fully charged one. This introduces scenarios with the possibility of missing data such as if a CGM scan is missed, the ECG device is not changed when the battery is depleted or it is not correctly activated. To mitigate this, participants will be provided with instructions and a demonstration as well as an information sheet. A member of the research team will also be contactable throughout if further assistance is needed.

To our knowledge, no studies have attempted to develop a glycaemic event detection model for people with T1DM using deep learning with raw single-lead ECG signals. This study is being run complementary and in parallel to another involving paediatrics with T1DM.[50] In our planned analysis, we will investigate changes in the ECG morphology and other features derived from the ECG signal with respect to glycaemic status. A novel aspect of the analysis will be the inclusion of additional features such as activity levels, mealtimes and composition, and sleep patterns.

Non-invasive monitoring of blood glucose and glycaemic event detection can potentially eliminate the need for finger-pricking. The development of an accurate and robust model for the non-invasive detection of abnormal excursions of blood glucose is pivotal for efficient management of metabolic disorders such as diabetes, drastically reducing discomfort, costs and waste associated with current invasive methods to measure blood glucose concentration. A continuous non-invasive blood glucose monitoring solution could help to overcome barriers and limitations of traditional methods, thereby increasing adherence to self-management protocols leading to improved outcomes, quality of life and reduced incidence of complications.

## Ethics and dissemination

This study has received ethical approval from the Research Ethics Service (ref: 17/NW/0277). It is anticipated that the scientific findings of the study will be disseminated via presentation at national or international conferences and through publication in peer-reviewed scientific journals.

**Author affiliations**
[1]School of Engineering, University of Warwick, Coventry, UK

[2]Division of Biomedical Sciences, Warwick Medical School, University of Warwick, Coventry, UK

[3]Warwickshire Institute for the Study of Diabetes, Endocrinology and Metabolism, University Hospitals Coventry and Warwickshire NHS Trust, Coventry, UK

[4]Human Metabolism Research Unit, University Hospitals Coventry and Warwickshire NHS Trust, Coventry, UK

[5]Department of Engineering, Università Campus Bio-Medico di Roma, Rome, Italy

**Contributors** LP and JH conceived the idea. OC, KS, BO, MSH, JH and LP contributed to the development of the protocol, study design and methods. OC wrote the first draft. KS, BO, MSH, TMB, MOW, JH and LP critically revised the draft. All authors have approved the final written manuscript.

**Funding** This study is funded by the Warwick-Wellcome Trust Translational Partnership Award and the Engineering and Physical Sciences Research Council Impact Acceleration Account (EPSRC IAA). KS is funded by the MRC Doctoral Training Partnership (grant number MR/N014294/1).

**Competing interests** None declared.

**Patient and public involvement** Patients and/or the public were not involved in the design, or conduct, or reporting, or dissemination plans of this research.

**Patient consent for publication** Not required.

**Provenance and peer review** Not commissioned; externally peer reviewed.

**ORCID iDs**
Owain Cisuelo http://orcid.org/0000-0002-0967-9772
Katy Stokes http://orcid.org/0000-0003-0766-6836
Muhammad Salman Haleem http://orcid.org/0000-0001-5946-6567

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
