## [Reviewer comments · BMJ Open]

ARTICLE DETAILS

TITLE (PROVISIONAL)	Development of an artificial intelligence system to identify hypoglycaemia via ECG in adults with type 1 diabetes: protocol for data collection under controlled and free-living conditions
AUTHORS	Cisuelo, Owain; Stokes, Katherine; Oronti, Busola; Haleem, Salman; Barber, Thomas; Weickert, Martin; Pecchia, Leandro; Hattersley, John

VERSION 1 – REVIEW

REVIEWER	Saez, Marc Universitat de Girona, Research Group on Statistics, Econometrics and Health (GRECS)
REVIEW RETURNED	03-Oct-2022

GENERAL COMMENTS	The authors attempted to provide a methodological description of an observational study in which we aim to obtain physiological data from people with T1DM to validate and improve the AI for use in this population. They also intended to investigate a general AI model for glycemic event detection. The authors have been quite successful in achieving their objectives. However, I have a couple of comments, which I would consider major. 1.- Have the authors not thought to consider covariates that could be related to the association of interest? Authors should explain in some detail which covariates they consider and if they don't, why not. 2.- Authors should explain, also in some detail, the limitations of their paper. They should explain how these would influence the results, as well as how they could be avoided. In the case of not being able to avoid them, because they could not do it
--

REVIEWER	Yan, Xiaoxi Duke-NUS Medical School, Centre for Quantitative Medicine
REVIEW RETURNED	15-Dec-2022

GENERAL COMMENTS	The protocol aims to conduct an observational study to collect physiological data to develop an AI model for predicting hypoglycaemia in T1DM patients and develop an AI model for glycaemic event detection. The paper is concise and generally well-written. Below are my specific comments. 1. Could the authors provide further justification as to how the sample size (approximately 30) is sufficient for both the development and validation of an AI model for a more heterogeneous population (as compared to the previous work (reference 15), where it was only
---

	for healthy elderly males)? The authors also mentioned in the Discussion section (p.8, lines 28-30) that “The study will add to the body of evidence.... Current efforts in the literature have been limited by small sample sizes,...”. Although the study does generate data from T1DM patients in free-living conditions, I would say the sample size is still relatively small; in other words, it is not very different from the other studies in the literature. It may be better to rephrase the sentences so that the readers are clear about where exactly the study overcomes the limitations in the literature. 2. The Outcomes section (p.7) reads more like a description of the aims and some data modelling approaches. Given the study context, I am unsure if there’s a need for an Outcome section; in my opinion, most of the paragraphs would fit better under an Aims/Objectives section. 3. I am confused about whether the “general glycaemic event detection algorithm” (p.7, lines 43-44) is a secondary outcome or part of the primary outcome. Because in the Discussion (p.8, line 22), it says that it is a secondary outcome. If true, the sentence (p.7, lines 43-44) should be moved under the Secondary Outcome section to avoid confusion. 4. In addition, could the authors clarify if the “general glycaemic event detection algorithm” can be used in “diabetic and healthy subjects” as described in the Discussion (p.8, line23) or only in “diabetic subjects” as described in the Abstract Introduction (p.3, line 13-15)? 5. Will there be scenarios where the patients are considered non-compliant? And if yes, how will the research team deal with non-compliance? 6. I think the authors can include more limitations of the study in the Discussion. 7. I also spotted some writing errors; it would be great if the authors could carefully review the manuscript. Thank you!
--	---

VERSION 1 – AUTHOR RESPONSE

Reviewer 1		
R1.1	Have the authors not thought to consider covariates that could be related to the association of interest? Authors should explain in some detail which covariates they consider and if they don't, why not.	We thank the reviewer for raising this issue. Details of the potential covariates to consider and how we may handle them in our analysis has been included in the analysis section on page 6. 'The autonomic nervous system is

		responsible for maintaining homeostasis and regulates processes such as blood pressure, digestion, metabolism, and circadian rhythm, all of which may affect cardiac function and show as changes in the ECG morphology. Additionally, exercise can affect the QT interval, that is the ECG section representing ventricular depolarization, due to exercise-induced autonomic response. During phase 1 of the study daytime blood pressure will be monitored, and the calorific value and macronutrient composition of meals will be recorded. Additionally, exercise sessions will be scheduled throughout the phase. Throughout the free-living phase, all meals, exercise duration and intensity, and sleep times will be recorded by the participant. The data collected via this protocol enables us to examine how the covariates relate to the outcome of interest, i.e., the glycaemic status by performing statistical tests such as analysis of variance (ANOVA). The recording of mealtimes, exercise, and sleep enables the data to be analysed during distinct periods of interest such as post-exercise or postprandial.'
R1.2	Authors should explain, also in some detail, the limitations of their paper. They should explain how these would influence the results, as well as how they could be avoided. In the case of not being able to avoid them, because they could not do it	The discussion section has been expanded to consider some limitations of the study design. 'A potential limitation of this protocol is that during the free-living phase of the study, participants will be required to extract data from the CGM at least once every eight hours. They will also have to remove and reattach the ECG device when engaging in water-based activities or exchanging the device for a new fully charged one. This introduces scenarios with the possibility of missing data such as if a CGM scan is missed, the ECG device is not changed when the battery is depleted, or it is not correctly activated. To mitigate this, participants will be provided with instructions and a demonstration as well as an information sheet. A member of the research team will also be contactable throughout if further assistance is needed.'

Reviewer 2

R2.1	Could the authors provide further justification as to how the sample size (approximately 30) is sufficient for both the development and validation of an AI model for a more heterogenous population (as compared to the previous work (reference 15), where it was only for healthy elderly males)?	The Sample size section has been expanded to highlight that in the context of this study the sample size is with respect to the number of observations and not number of participants. ‘Full participation in the protocol will yield 108 hours of continuous data per participant. We aim to capture the daily blood glucose fluctuations known as glycaemic variability, a feature of impaired glucose metabolism. However, it is not possible to anticipate how many hypoglycaemic episodes will be recorded. ECG data, labelled with corresponding glucose concentration, will be analysed in excerpts of variable length from individual heart beats (cardiac cycles) to several minutes, resulting in a large sample size.’
	The authors also mentioned in the Discussion section (p.8, lines 28-30) that “The study will add to the body of evidence.... Current efforts in the literature have been limited by small sample sizes,...”. Although the study does generate data from T1DM patients in free-living conditions, I would say the sample size is still relatively small; in other words, it is not very different from the other studies in the literature. It may be better to rephrase the sentences so that the readers are clear about where exactly the study overcomes the limitations in the literature.	We thank the reviewer for highlighting this point for clarification. The aspects of the study that overcome limitations of the existing literature have now been highlighted in the discussion. ‘This study protocol has been developed to address the gaps of the existing literature. As such, the strengths of this study design are that data is collected from participants with diabetes, both in controlled and free-living settings. An additional strength of this protocol is that it has been designed with non-compliance in mind. The first phase will take place in a metabolic chamber where the activities of the participants and the operation of the wearable sensors are prescriptive and supervised. This enables us to obtain high quality baseline data. The second phase will take place in free-living conditions where the participants go about their normal daily activities unencumbered by the wearable sensors, providing real-life data. During the free-living phase a level of non-compliance and variance is desirable in order to determine how our analysis and

		model works with real-world data. Non-compliance with the protocol will not exclude the participant from inclusion in the analysis. Any deviations from the protocol will be noted. Participation in both phases of the study is not a requirement, therefore we are likely to obtain different number of samples for each participant. We will take a pragmatic approach and work with the data available at end of the study. All analysis will report the number of samples used and if excerpts of data have been excluded and for what reason. Our planned analysis will segment the continuous physiological data into smaller excerpts of variable length, yielding a sufficient sample size for the training algorithm to discover relevant features in the data.'
R2.2	The Outcomes section (p.7) reads more like a description of the aims and some data modelling approaches. Given the study context, I am unsure if there's a need for an Outcome section; in my opinion, most of the paragraphs would fit better under an Aims/Objectives section.	Thank you for this observation. The Objectives section on page 3 has been clarified and some points from the outcomes have been incorporated. Table 2, which was previously in the outcome section has been moved to data analysis on page 6 Study Objectives The purpose of this observational study is to acquire continuous physiological data from adults with T1DM over a period of several days. Data obtained from wearable sensors and recorded in diaries to track lifestyle activities will be used to carry out our primary and secondary objectives. Primary Objective The primary objective of this study is to further develop and validate in a new population (T1DM) our previously developed AI approaches for non-invasive hypoglycaemia detection. We will extract and analyse ECG-derived features from

		the raw signal and determine their relationship with the glycaemic status. State-of-the-art algorithms will be used to create a mathematical mapping from the ECG-derived features to the glycaemic status. Secondary Objective To examine the impact of physical activity and diet on glycaemic events and incorporate relevant features into the model.
R2.3	I am confused about whether the “general glycaemic event detection algorithm” (p.7, lines 43-44) is a secondary outcome or part of the primary outcome. Because in the Discussion (p.8, line 22), it says that it is a secondary outcome. If true, the sentence (p.7, lines 43-44) should be moved under the Secondary Outcome section to avoid confusion.	We thank the reviewer for highlighting this area of confusion. We have adjusted the language used to help make our objectives clearer and consistent. The abstract, discussion and objectives have been adjusted and aligned: Page 2: Abstract: ‘The aim of this work is to further improve the previously developed AI model and validate its performance for glycaemic event detection in people with T1DM. ‘ Page 3: Objectives: ‘The purpose of this observational study is to acquire continuous physiological data from adults with T1DM over a period of several days. Data obtained from wearable sensors and recorded in diaries to track lifestyle activities will be used to carry out our primary and secondary objectives. Primary Objective The primary objective of this study is to further develop and validate in a new population (T1DM) our previously developed AI approaches for non-invasive hypoglycaemia detection. We will extract and analyse ECG-derived features from the raw signal and determine their relationship with the glycaemic status. State-of-the-art algorithms will be used to

		create a mathematical mapping from the ECG-derived features to the glycaemic status. Secondary Objective To examine the impact of physical activity and diet on glycaemic events and incorporate relevant features into the model.' Page 7: Discussion: 'The primary objective of this observational study is to obtain physiological data from people with T1DM using non-invasive wearable sensors to validate and further develop an AI model for automated detection of hypoglycaemia.'
R2.4	In addition, could the authors clarify if the "general glycaemic event detection algorithm" can be used in "diabetic and healthy subjects" as described in the Discussion (p.8, line23) or only in "diabetic subjects" as described in the Abstract Introduction (p.3, line 13-15)?	We have addressed this point in response to R2.3.
R2.5	Will there be scenarios where the patients are considered non-compliant? And if yes, how will the research team deal with non-compliance?	Thank you for this comment. Participant non-compliance with the protocol and how we will deal with it has been added to the discussion. 'During the free-living phase a level of non-compliance and variance is desirable in order to determine how our analysis and model works with real-world data. Non-compliance with the protocol will not exclude the participant from inclusion in the analysis. Any deviations from the protocol will be noted. Participation in both phases of the study is not a requirement, therefore we are likely to obtain different number of samples for each participant. We will take a pragmatic approach and work with the data available at end of the study. All analysis will report the number of samples used and if excerpts of data have been

		excluded and for what reason'
R2.6	I think the authors can include more limitations of the study in the Discussion.	The discussion section has been expanded to consider some limitations of the study design. Please see R1.2.
R2.7	I also spotted some writing errors; it would be great if the authors could carefully review the manuscript.	We thank the reviewer for highlighting this, the manuscript has been closely read to pick up any errors. All changes are highlighted in yellow.

VERSION 2 – REVIEW

REVIEWER	Saez, Marc Universitat de Girona, Research Group on Statistics, Econometrics and Health (GRECS)
REVIEW RETURNED	26-Feb-2023

GENERAL COMMENTS	The authors have answered very well not only to my comments, but also those of the other reviewers. In addition, many have been included in the new version of the manuscript. I have no further comments.
--

REVIEWER	Yan, Xiaoxi Duke-NUS Medical School, Centre for Quantitative Medicine
REVIEW RETURNED	06-Mar-2023

GENERAL COMMENTS	The authors have satisfactorily addressed my comments. I have no further suggestions.
---